# Triblock Copolymer Compatibilizers for Enhancing the Mechanical Properties of a Renewable Bio-Polymer

**DOI:** 10.3390/polym14132734

**Published:** 2022-07-04

**Authors:** Guilian Xue, Bohua Sun, Lu Han, Baichuan Liu, Hongyu Liang, Yongfeng Pu, Hongming Tang, Fangwu Ma

**Affiliations:** 1State Key Laboratory of Automotive Simulation and Control, College of Automotive Engineering, Jilin University, Changchun 130022, China; xuegl20@mails.jlu.edu.cn (G.X.); liubc1709@sina.com (B.L.); lianghy19@mails.jlu.edu.cn (H.L.); puyongfeng@jlu.edu.cn (Y.P.); mikema@jlu.edu.cn (F.M.); 2Changguang Jizhi Optical Technology Co., Ltd., Changchun 130022, China; hanlu.jay@163.com; 3Key Laboratory of Polymer Ecomaterials, Changchun Institute of Applied Chemistry, Chinese Academy of Sciences, Changchun 130022, China; kqli@ciac.ac.cn

**Keywords:** triblock copolymer, compatibilization, bioplastic, reinforced composite, failure analysis

## Abstract

Poly(lactic acid) (PLA) is an emerging plastic that has insufficient properties (e.g., it is too brittle) for widespread commercial use. Previous research results have shown that the strength and toughness of basalt fiber reinforced PLA composites (PLA/BF) still need to be improved. To address this limitation, this study aimed to obtain an effective compatibilizer for PLA/BF. Melt-blending of poly(butylene adipate-co-terephthalate) (PBAT) with PLA in the presence of 4,4′-methylene diphenyl diisocyanate (MDI: 0.5 wt% of the total resin) afforded PLA/PBAT-MDI triblock copolymers. The triblock copolymers were melt-blended to improve the interfacial adhesion of PLA/BF and thus obtain excellent performance of the PLA-ternary polymers. This work presents the first investigation on the effects of PLA/PBAT-MDI triblock copolymers as compatibilizers for PLA/BF blends. The resultant mechanics, the morphology, interface, crystallinity, and thermal stability of the PLA-bio polymers were comprehensively examined via standard characterization techniques. The crystallinity of the PLA-ternary polymers was as high as 43.6%, 1.44× that of PLA/BF, and 163.5% higher than that of pure PLA. The stored energy of the PLA-ternary polymers reached 20,306.2 MPa, 5.5× than that of PLA/BF, and 18.6× of pure PLA. Moreover, the fatigue life of the PLA-ternary polymers was substantially improved, 5.85× than that of PLA/PBAT-MDI triblock copolymers. Thus, the PLA/PBAT-MDI triblock copolymers are compatibilizers that improve the mechanical properties of PLA/BF.

## 1. Introduction

Poly(lactic acid) (PLA) is renewable and biodegradable, and exhibits many useful properties (such as good biocompatibility and extreme mechanical strength) that might help researchers replace petroleum-based materials [1,2]. Thus, it is prominent in the emerging bio-plastics market [3,4]. However, PLA is insufficient for most commercial applications, especially in engineering fields such as the automotive industry, because of its innate shortcomings (such as brittleness, slow rate of crystallization, and hydrolysis [5,6]). The mechanical performance of polymers depends on several parameters such as the enhance phase, interface compatibility and the preparation technology [7,8,9]. Polymer melt-blending is an effective and economical means of improving polymer properties.

Researchers have investigated biodegradable fibers as reinforced composites that exhibit superior performance and suitability for widespread use. Such composites combine the properties of the matrix polymer and reinforced fiber, and are lightweight and high strength [10,11,12]. The matrix material and reinforced fiber must meet the needs of engineering. For example, Liang et al. [13] used a mechanical stripping method, added 0.5 wt% graphene oxide and 9 wt% carbon fiber, and achieved mechanical enhancement. Azlin et al. [14] enhanced the thermal stability and degradation of PLA by adding a woven polyester fiber; the composites have potential for use in the automotive component industry. Jang [15] fabricated BF (basalt fiber) infiltrated polycarbonate composites; the mechanical robustness and rheology were examined via precise control of the BF content (up to 12.5 phr). Zaghloul et al. [16] prepared glass fiber-reinforced polyester specimens containing cellulose nanocrystals by using the optimum incorporation percentage of cellulose nanocrystals, and the addition of 4% cellulose nanocrystals to the polyester matrix led to the optimum tensile and fatigue properties. In previous research, our team fabricated BF melt-blended with PLA, and the tensile properties were well-improved via precise control of the BF content.

Nonetheless, the mechanical properties and toughness of the reinforced composites require improvement (e.g., the toughness). Thus, researchers have carried out extensive corresponding studies [17]. For example, inorganic materials [18,19,20]; compatibilizers [21,22,23]; plasticizers [24,25]; and copolymers [26,27] have all been used to toughen reinforced composites. In particular, the melt-blending of biomaterials is an effective approach. Many biodegradable polymers are applicable such as poly (butylene adipate-co-terephthalate), for example, Nomadolo et al. [28] suggested that industrial composting conditions are the most suitable for enhanced biodegradation of biopolymer blends such as: poly (butylene adipate terephthalate) (PBAT) blended with poly (lactic acid) (PLA), and PBAT blended with poly (butylene succinate) (PBS). Wang et al. [29] obtained fused filament fabrication with high modulus, flexibility, toughness and notched impact strength result upon combining PLA/PBAT (80:20; mass basis) with a sufficiently high cloister amount (1–5 phr) supported by a suited Joncryl amount (0.3 phr); aliphatic polyester poly(ε-caprolactone), for example, Voorde et al. [30] explored the fabrication of blended poly(lactic acid)/poly(ε-caprolactone) (PLA/PCL) fibers manufactured through multilayer coextrusion. Fenni et al. [31] established a clear relationship between the unique partial wetting morphology of ternary blends and the nucleation of the minor component by melt-blending PLA, PCL, and PBS to produce a “partial wetting” morphology. Priselac et al. [32] proposed a biodegradable blend of poly(ε-caprolactone) (PCL) and poly(lactic acid) (PLA) for the production of relief printing plates for special applications in packaging materials. Other polymers such as poly(butylene succinate) [33,34], poly(ethylene oxide) [35,36], and poly(ethylene glycol) [37,38] have also been applied. Blending resulted in the superior performance of the corresponding reinforced composites. Furthermore, a means of improving the interfacial adhesion of the composites is critical to obtaining good performance [39,40]. This work presents the first investigation on the effects of PLA/PBAT-MDI triblock copolymers as compatibilizers for PLA/BF blends to improve the interfacial adhesion of PLA-bio polymers, and the addition of 60% triblock copolymers to the PLA-bio polymers led to the optimum tensile and fatigue properties.

In this paper, melt-blending was used as the processing platform to manufacture PLA blends. The procedure was as follows. (1) PLA/PBAT binary polymers exhibited improved interfacial compatibility by addition of the compatibilizer 4,4′-methylene diphenyl diisocyanate (MDI). (2) The mechanism of fiber reinforcement by melting BF with PLA and PLA/PBAT-MDI triblock copolymers were revealed. (3) The means by which PBAT improved the mechanical properties of the PLA/BF blends was investigated. (4) The polymerization scheme and failure mechanism of the PLA ternary polymers was investigated. A series of PLA biopolymers was studied by varying the composition and a wide range of mechanical responses was obtained. The formulations (weight percent) of the copolymers in this article were set as follows: group A was PLA_n_/BF_m_ (n/m: 90/10, 80/20, 70/30, 60/40, 50/50, and 40/60 as A1–A5, respectively); group B was (10% PBAT/90% PLA) (B1), (10%PBAT/90% PLA-0.5% MDI) (B2), and (20% PBAT/80% PLA-0.5% MDI) (B3); and group C was (10% PBAT/90% PLA-0.5% MDI)_e_/BF_f_ (e/f: 90/10, 80/20, 70/30, 60/40, and 50/50 as C1–C5, respectively). Some of the copolymers exhibited poor mechanical properties, and thus were not discussed in detail. The connections between the morphology, crystallinity, and polymer mechanics were detailed, and the nanoscale characterizations, in terms of the morphologies and physicochemical properties, were used to understand the corresponding behavior at the macroscale. Polymers that exhibit excellent strength and toughness mechanics are suitable for applications in vehicle engineering.

## 2. Experimental

### 2.1. Materials

PLA (grade 290D, 1.85 g/cm^3^) and PBAT (1.22 g/cm^3^) were purchased from Zhejiang Haizheng Pharmaceutical (Zhejiang, China). BF (short-staple, 5–7 mm, 2.5–2.65 g/cm^3^) was purchased from Jilin Tongxin Basalt Technology (Tonghua, China) to strengthen the matrix component. All of the reagents were of technical or analytical grade, and used as received without further purification.

### 2.2. Preparation of Composites

PLA, PBAT, and BF were dried under vacuum to remove excess water for 48 h at 70 °C, 24 h at 30 °C, 24 h at 110 °C, respectively. Group A was composed of PLA-based binary blends that were melted with various proportions of BF. Groups B and C were PLA-based ternary blends that were melted with controlled morphology by using PBAT, MDI, and BF. Figure 1 is the design scheme for preparing high-performance, biodegradable, bio-based composites in this paper. The procedure was as follows. PLA was first blended with PBAT at 185 °C in a torque rheometer (XSS-300, Shanghai Kechuang rubber & Plastic Machinery Equipment Co., Ltd., Shanghai, China) with (or without) the reactive chain extender MDI (0.50 wt% of the total resin). Then, BF was melt-blended at 185 °C in accordance with five formulations. The temperature of the torque rheometer was set to 185 °C and the residence time was 10 min. Thus, the PLA, PBAT, MDI, and BF were uniformly mixed. Specimens were molded with an injection machine compression-molded sheet into a dumbbell shape to 25 mm × 4 mm × 2 mm for the uniaxial tensile and fatigue tests. Table 1 presents the properties of PLA, PBAT, and BF.

### 2.3. Characterizations

The morphology of the polymer architecture and fracture morphology were observed by scanning electron microscopy (SEM: LYRA3-XM, TESCAN, Brno, Czech Republic) at an accelerating voltage of 3 kV. The polymer samples were polished before mounting onto a sample stage and sputter-coated with a layer of Au/Pd prior to imaging. The wide-angle X-ray scattering (WAXS) data were collected with a D8 Advance detector from Bruker (BRUKER, Karlsruhe, Germany) at a pixel resolution of 486 px × 618 px (1 pixel = 0.172 cm) to collect the scattered beam. Fourier transform infrared spectroscopy (FTIR) was performed with dried samples by the attenuated total reflectance method with a spectrometer (BRUKER, Karlsruhe, Germany, HYRERION 3000), equipped with a high-sensitivity deuterated triglycine sulfate and mercury cadmium telluride detector accessory that incorporated a diamond internal reflection element. Spectra were collected for all of the samples at a resolution of 0.5 cm^−1^ in the range of 370–4000 cm^−1^ for a total of 32 scans, and were not corrected for penetration depth variation.

The thermal stability of the copolymer was characterized by dynamic thermo-mechanical analysis (DMA; NETZSCH-242E Artemis, NETZSCH, Bavaria, Germany) with a double cantilever in forced non-resonance mode at a heating rate of 2 °C/min, a test frequency of 1 Hz, and an amplitude of 20 μm under a liquid nitrogen flow from −40 °C to 200 °C. The thermal behavior of the copolymers was examined by dynamic scanning calorimetry (DSC; Perkin-Elmer Thermal Analysis, Perkin-Elmer, Norwalk, CT, USA). All of the copolymers, under a nitrogen atmosphere, were sealed in hermetic pans to determine the thermal transitions with a blank hermetic pan as a reference. The samples were initially cooled and held at −40 °C for 3 min to equilibrate, then heated to 200 °C at 10 °C/min, and then maintained for 5 min to eliminate the heat history before cooling down to −40 °C at 10 °C/min. These steps were repeated for the second heating.

The drawing splines were molded by an injection molding process with a HAAKE Mini Jet Pro injection machine (Thermo Scientific, Thermo Fisher Scientific, Waltham, MA, USA). The injection and mold temperature were set at 205 °C and 80 °C, respectively. The mixed composite materials were filled into the raw material chamber and heated for 10 s to melt the raw materials. Then, the injection pressure was set to 450 bar; the melted raw materials were then injected into the mold, pressed for 15 s at 150 bar in the mold, and then removed.

The uniaxial tensile tests were carried out at room temperature with an MTS Landmark 370–250 kN mechanical testing instrument (MTS Systems, Eden Prairie, MN, USA) at a 100 N load. Each specimen was tested a minimum of five times per sample. All of the tension–tension fatigue tests were operated under constant amplitude loading at a frequency of 2 Hz and a stress ratio R = 0.1 at room temperature. Tests were conducted with an MTS Landmark at 250–370 kN mechanical testing, in which the static load error for the full measurement range was ±0.2% and the dynamic load error was ±2%. At least five specimens were used to obtain each point of the applied stress–fatigue life (*S*–*N*) curve.

## 3. Results and Discussion

### 3.1. Structural Characterization of PLA-Based Binary and Ternary Blends

Structural characterizations of PLA-based binary and ternary blends (Figure 2 and Figure 3) were determined by SEM, WAXS, and FTIR.

Figure 2 shows the structural characterizations of the PLA-based binary and ternary blends in terms of the diameter and length of the BF, the surfaces, and cross-sectional SEM micrographs and the EDS of C2. Figure 2a indicates that the distribution of the BF diameter was 10.5–13.5 μm, in accordance with a positive skewness distribution. Figure 2c shows the surface SEM micrographs of C2. Figure 2b,d shows the lengths of the BF and cross-sectional SEM micrographs of C2, clearly indicating the BF length. The average fiber length of BF after compounding, as aforementioned, was 5–7 mm by default. Figure 2f,g is the EDS from the SEM micrograph of Figure 2d, where it can be clearly seen that Figure 2f is the EDS of BF with the elemental contents of (At%): 42.6% C, 35.5% O, 9.3% Si, 3.6% Al, 1.9% Mg, 1.7% Ca, 1.5% Fe, and 1.5% Na; the elemental content of Au may be affected by gold spraying. Figure 2g is the EDS of the matrix with the elemental contents of (At%): 60.9% C, 34.3% O, 3.4% Si; the elemental content of Si may be affected by the debris of BF after compounding. Figure 2e is the tensile fracture morphology of C2. Figure 2h,i shows the EDS from the SEM micrograph in Figure 2e. Figure 2h is the EDS of the matrix with the elemental contents of (At%): 59.8% Si and 40.2% Au; the element content of Au may be affected by gold spraying and Si may be affected by the debris of BF after compounding. Figure 2i is the EDS of BF with the elemental contents of (At%): 59.3% O, 16.3% Si, 12.5% Fe, 5.2% Al, 5.1% Ca, and 1.6% Au; the elemental content of Au may be affected by gold spraying.

The SEM micrographs of the polymers indicate that the BF melted homogeneously within the PLA/PBAT-MDI triblock copolymers. To obtain more intuitive surface and cross-sectional SEM micrographs, the PLA-bio polymer samples were polished before SEM, thus damage to the surface and cross-sections of the polymers were attributable to polishing and inevitable. A comprehensive comparison of Figure 2f–i shows that polishing did not affect the composition of the PLA-bio polymer samples.

The chemical structure of the polymers was confirmed by WAXS and FTIR (Figure 3). Group C was selected as the analysis group to observe the relationship between the chemical structure of the polymers and the material ratio. The peak, WAXS lines, and peak width of group C (Figure 3a) were similar, which indicates that no new diffraction peaks were discerned after BF was added. One can determine the crystallinity of the material by observing the full width at half maximum (FWHM) in the WAXS of group C. Accordingly, adding BF sharpened the diffraction peak; thus adding BF increased the crystallinity of the polymer.

Five bonds can be found in PLA (Figure 3c) [41]: C–C stretching at 867 cm^–1^; C–O stretching at 1079, 1126, 1181, and 1264 cm^–1^; C–H deformation at 1377 and 1452 cm^–1^, C=O ester carbonyl groups at 1748 cm^–1^; and C–H stretching at 2853 cm^–1^ and 2924 cm^–1^. Figure 3b shows the FTIR results: the assigned peaks of group C, their characteristic bonds, and the difference in intensity and width of the peaks were all analogous with PLA. In particular, the results indicate that the C–O and C–C bonds form the PLA backbone, which can be degraded by sunlight. This may confirm that the addition of BF and PBAT does not destroy the basic skeleton of PLA. Three different areas were evident in Figure 3b. The range of 3150–3400 cm^−1^ showed two faint peaks that corresponded to N–H and gradually weakened in accordance with the increasing BF content, and even tended to be parallel to the x-coordinate axis. C1–C4 showed two weak peaks from 1000–1050 cm^−1^, whereas B2 showed only one peak within the range, as seen in Figure 3d. The literature indicates that silicon-based polymers were evident at 1000–1050 cm^−1^ [42,43], thus the difference in the purple area of Figure 3d might be due to the silicon-based materials in the BF.

### 3.2. Mechanical Properties of PLA-Based Binary and Ternary Blends

The mechanical properties of the PLA-based binary and ternary blends (Figure 4, Figure 5, Figure 6 and Figure 7) were evaluated in terms of the thermal endurance, tensile, and tension–tension fatigue properties.

Regarding the thermal endurance properties, the glass transition temperature (*T_g_*) was equal to half the extrapolated C_p_; the melting temperature (*T_m_*) was measured at the endothermic peak; and the crystallization temperature (*T_c_*) was measured at the exothermic peak. *T_cc_* is the crystallization temperature in the cooling cycle. The percent crystallinity (*X_c_*) of the polymers was calculated from the DSC melting enthalpies (Equation (1)) using a value of 93 J/g for 100% crystalline PLA. Additionally, the weight percent (*w*) was included in the calculations of the crystallinity and cold crystallization enthalpy (Δ*H_cc_*) for PLA.
(1)$$X_{c}=\frac{\Delta H_{f}-\Delta H_{cc}}{w\Delta H_{f}^{o}}$$

The thermal behavior of groups A and C were examined by DSC (Figure 4a,b), where Figure 4c indicates the changing trend of *X_c_* between groups A−C. Figure 4a indicates that BF had little effect on the crystallization starting and ending temperature of group A. The initial crystallization temperature of group A ranged from 103 °C to 105 °C, and the end temperature was ca. 108 °C. The exothermic enthalpy of the PLA-bio polymers was calculated by integration, and the value of group A decreased from 46.37 J/g (A1) to a value of 18.35 J/g (A6), while the value of group C decreased from 42.374 J/g (C1) to 28.817 J/g (C4). This occurred when the material crystallized gradually decreased; the uncrystallized part of the material decreased with increasing BF content, and thus BF played an auxiliary role in the crystallization behavior.

The initial crystallization temperature of group C (Figure 4b) ranged from 99 °C (B2) to 103.47 °C (C4). Thus, C1–C4 exhibited a higher glass transformation temperature (*T_g_*): at least 2.89× that of B2. Furthermore, there was no variation in the *T_g_* among C1–C4 in the blends. The *X_c_* of groups A–C (Figure 4c) changed from 63.5% (A1) to 91.18% (C4), thus the *X_c_* of groups A–C was at least 83.5% (A1) higher and 163.5% higher (C4) than that of PLA; these all show that the addition of PBAT and BF have played an important role in the increase in the *X_c_*.

The mechanical properties of the PLA biopolymer binary and ternary blends as well as the thermal endurance properties (Figure 5) were analyzed by DMA. The thermal endurance properties of the PLA biopolymers (storage moduli in Figure 5a,c, and the tanδ values in Figure 5b,d) indicate that BF positively influenced the storage modulus and could delay the degradation of PLA. A polymer blend with a higher BF content can still exhibit a higher elastic modulus.

The storage modules of composites with various BF contents (Figure 5a,c indicates that the stored energies of A6, C2, and C4 reached 3117.5, 9456.55, and 20,306.2 MPa, respectively, at 30 °C, which were 3×, 9×, and 19.6× that of pure PLA (1035.5 MPa), respectively, at 30 °C. These results might be due to the fact that A6, C2, and C4 exhibited excellent structural strength and stiffness at this temperature. The PLA matrix softened gradually and the storage modulus of the polymer decreased slightly with increasing temperature to 25 °C, whereas the storage energy of A6 decreased by 2%, and the other polymers remained unchanged (which is consistent with −40 °C). The storage modulus of groups A and C decreased substantially from 45 °C to 70 °C, which might be attributable to BF, hindering the macromolecule secular chain movement of PLA. The storage modulus of the polymers first decreased gradually, but substantially when approaching the *T_g_* due to the vitrification transformation of PLA. The increase in temperature of the tanδ curves of group A (Figure 5b) and group C (Figure 5d) was between 56 °C and 60 °C, which is consistent with the *T_g_* measured by DSC (Figure 4). The mechanical loss of the polymers increased, reaching a maximum when the temperature reached 70 °C, and the energy storage modulus E’ decreased to a minimum in accordance with increasing temperature, after which the polymers entered a highly elastic state.

Certain applications use materials as structural parts over temperatures from −40 °C to *T_g_* (58 ± 2 °C) (i.e., the nonmetallic decoration and structural parts of automobiles). The increase in the crystallinity and storage modulus of groups A and C in the DMA test indicates improved mechanical properties compared with those of PLA, and the stability of the relationship between the mass fraction and tensile properties is important for design and performance predictions in future work.

The tensile properties of the polymers (Figure 6) indicate the effect of adding BF and the mesophase PBAT into the PLA matrix. The elongation at break (Figure 6a), tensile strength (Figure 6b), and tensile modulus (Figure 6c) indicate that compared with PLA, regarding polymers of groups A and C, the tensile modulus and tensile strength of group B decreased because PBAT improved the tenacity of PLA, whereas the elongation at break was not substantially improved. The tensile strength of the polymers gradually increased with the content of BF, and it should be noted that the tensile strength of group A was higher than that of group C at the same BF content, and the tensile strength of the polymers gradually increased with an increasing content of BF. Conversely, the tensile modulus increased from 3.2 GPa (PLA) to 8.6 GPa (C4), and the tensile modulus of C4 was 168.8% higher than PLA. These indicate that BF played an important role in enhancing the tensile properties of PLA.

BF can increase the mechanical and thermal properties of PLA, but the toughness of the material requires further improvement. The main problem lies in the brittleness of PLA as a matrix. Over the lifetime of automobile products, fatigue performance is also an important factor that affects the product quality.

The *S*–*N* curve indicates the relationship between the applied stress and fatigue life, from which one can obtain the fatigue strength (which corresponds to a cyclic number). Tension–tension fatigue tests were only carried out on B1, B2, and C1–C4 because of the tensile performance test data from Figure 6. Five stress levels (75%, 65%, 50%, 40%, and 30%) of the static strength were set as the maximum of the classification and five effective data points for each stress level. The *S*–*N* curves and fatigue point of the polymers were cycled at constant amplitude until failure (Figure 7).

The *S*–*N* curve of C4 was the highest under five stress levels, the value of C2 was the second-highest in the experimental group, and the number of cycles in the same cycle stress of B1 and B2 was much lower than that of group C (Figure 7a). Figure 7b shows the fatigue point of the polymers. The data dispersion of the five groups of fatigue test points was small and within the confidence interval, thus the fatigue test data in this paper was precise.

### 3.3. Fracture Morphology of PLA-Based Binary and Ternary Blends

SEM was used to observe the tensile fracture and fatigue fracture morphology of the PLA-bio polymers. Figure 8 shows the results for PLA, A1, A2, A4, B2, and C1–C4. Figure 9, Figure 10 and Figure 11 show the fatigue fracture morphology of C1, C2, and C4. These experiments were performed to analyze the mechanical properties and reveal the fracture mechanism.

The impact tensile fracture surface of PLA, groups A, B, and C can be seen in Figure 8, and the phenomenon of area 1 in Figure 8 is craze, area 2 is fiber pull-out, area 3 is fiber fracture, and area 4 is fiber debonding. Figure 8 indicates a brittle morphology of PLA, and a toughness one for B2; craze and additional roughness were found in B2. A1 was dominated by craze, fiber pull-out, and fiber fracture. A2 had fiber pull-out as the main failure morphology. A4 exhibited fiber fracture and fiber pull-out. C1 was dominated by craze and fiber pull-out, and the fiber fracture occurred locally. The morphology of C2 exhibited fiber pull-out, craze, and fiber debonding. The tensile fracture of C3 exhibited craze, fiber fracture, fiber pull-out, and local fiber debonding. The fracture morphology of C4 indicated fiber fracture, and there was good interfacial compatibility between the BF and matrix after tensile fracture.

The fatigue fracture morphology was investigated by SEM; the fatigue fracture mechanism was determined by the fatigue *S*–*N* curve and the fatigue fracture morphology. The *S*–*N* curves of C1 and C3 exhibited the same trends and confirmed the aforementioned analysis. The fatigue fracture morphology (at five stress levels) of C1, C2, and C4 was analyzed by SEM (Figure 9, Figure 10 and Figure 11).

Figure 9 indicates the fracture morphology and fatigue point of C1 under five stress levels: 30%, 40%, 55%, 65%, and 75%.

Figure 9a–e indicates the fatigue fracture morphology as determined by SEM. Area 1 exhibited craze, area 2 exhibited fiber fracture, area 3 exhibited fiber pull-out, area 4 exhibited layered peeling, and area 5 exhibited fractured BF. Lamination might indicate a reduction in the fatigue life of C1 at stress levels of 55% to 75%. This result agrees with the *S*–*N* curve of C1 in Figure 7. Figure 9f indicates the fatigue point of C1; the data dispersion of the five groups of fatigue test points was small and within the confidence interval.

Figure 10 indicates the fracture morphology as determined by SEM and the fatigue point of C2. Figure 10a–e indicates the fracture morphology of C2 under the five aforementioned stress levels. The fracture morphology in Figure 10a,b was mainly fiber fracture and fiber pull-out; layered peeling was evident in Figure 10c–e. Figure 10f indicates the fatigue point of C2. The data dispersion of the five groups of fatigue test points was small and within the confidence interval, and the cyclic stress amplitude ([S-1]) of C2 was higher than that of C1 in Figure 9f, the same as the number of cycles (lgN).

Figure 11 indicates the fracture morphology as determined by SEM and the fatigue point of C4. Figure 11a,b indicates the fracture morphology of C4 under the five aforementioned stress levels. The fracture morphology in Figure 11a–c was mainly fiber fracture, whereas that in Figure 11d,e was mainly fiber fracture and fiber pull-out.

Unlike the morphology of C1 in Figure 9 and the morphology of C2 in Figure 10, C4 had a high fiber content, but no layered peeling phenomenon was observed in Figure 11. The fibers that remained in the matrix after fiber fracture were still tightly bound to the matrix. This phenomenon increased the fatigue life of C4, and can explain the phenomenon that the fatigue life of C4 in the *S*–*N* curve was the highest in Figure 7. Figure 11f indicates the fatigue point of C4. The data dispersion of the five groups of fatigue test points was within the confidence interval. Furthermore, the cyclic stress amplitude ([S-1]) and the number of cycles (lgN) of C4 were higher than those of C1 and C2. These results are consistent with the results indicated by the *S*–*N* curve.

The corresponding fatigue life of C4 at each stress level was higher than that of C1 and C2 at the same stress, as determined by comparing the fracture morphologies at the five stress levels from Figure 9, Figure 10 and Figure 11, and the fatigue *S*–*N* curve in Figure 7. Thus, at the same stress level, the fatigue life of C4 was the highest, followed by C2, and C1 was the lowest. In accordance with the fracture morphology, the matrix and fiber bonds of C4 and C2 were close under the five aforementioned stress levels, thus the close bond enhanced the fatigue strength of C4 and C2. The increased fatigue life of C4 might be due to the fiber strengthening effect of BF.

### 3.4. Fatigue Mechanical Analysis

The mechanism of microcosmic expansion is complex and various for the interaction and competition between silver striations and shear bands in polymer materials, and the force in the fatigue test is cyclic stress, so the effect of orientation on the failure properties can complicate the stress situation, accordingly.

Figure 12 shows the polymerization schemes and fatigue failure mechanism of the polymers. Figure 12A shows the polymerization scheme [44,45]. Figure 12B,a–e reveals the fatigue failure mechanism of the polymers, determined by the fatigue *S*–*N* curve and the fatigue fracture morphology. Figure 12A,B shows the schemes of the fatigue test and fracture morphology of the polymers in groups A and C. The fatigue fracture morphology in Figure 12b exhibited fiber pull-out, fiber debonding, fiber fracture, and fiber debonding, which correspond to the sequence between BF and the matrix.

The force status that was applied to the fatigue failure mechanism of the polymers was analyzed (Figure 12c–e) from the corresponding microscopic stress decomposition diagrams. The *f* in the red box is the friction between the fibers and matrix; the *f* in the yellow box is the friction from the fibers; and the overlap of the two cases above refers to the friction from the fibers and the friction between the fibers and matrix.

Different frictions can cause different fracture morphologies such as fiber pull-out, fiber debonding, or fiber fracture.

As an effective compatibilizer, PLA/PBAT-MDI triblock copolymers play an important role in improving the properties of PLA/BF. The PLA ternary polymer C4 exhibited synergistic mechanical properties with high extensibility. The crystallinity of C4 was as high as 43.6 %, 1.44× that of PLA/BF, and 163.5% higher than that of pure PLA. The stored energy of C4 reached 20,306.2 MPa, 5.5× than that of PLA/BF and 18.6× of pure PLA. Moreover, the fatigue life of the PLA-ternary polymers was substantially improved, 5.85× than that of the PLA/PBAT-MDI triblock copolymers. These were all based on comprehensive analyses such as DSC, DMA, tensile tests, fatigue tests, and the failure mechanism (Figure 4, Figure 5, Figure 6, Figure 7, Figure 8, Figure 9, Figure 10, Figure 11 and Figure 12).

## 4. Conclusions

In this work, three groups A (PLA/BF), B (PLA/PBAT, with or not with MDI), and C (PLA-PBAT-MDI/BF) with controlled morphology were investigated, based on PLA/PBAT-MDI triblock copolymers and BF. A series of experimental design schemes was studied by the variation in the composition range, obtaining a wide range of mechanical responses.

The structural characterization of the PLA-bio polymers was observed by SEM, WAXS, and FTIR. SEM demonstrated that the polymers have a great interface compatibility and mixing homogeneity. Similar WAXS lines indicated that no new diffraction peaks were evident after adding BF. The FTIR indicates that the characteristic bonds of the polymers were all analogous with those of PLA. In particular, the results indicate that the C–O and C–C bonds form the PLA backbone, which can be degraded by sunlight.

The thermal behavior of the PLA-bio binary and ternary polymers was examined by DSC and DMA. DSC analyses and the increase in the crystallinity and storage modulus in the DMA tests indicated that the triblock copolymers played an auxiliary role in the crystallization, and furthermore improved the mechanical properties and stability of the relationship between the mass fraction and tensile properties of the polymers. These are important for the design and performance predictions in future work.The tensile tests indicated that the tensile strength of the polymers gradually increased in accordance with the content of BF, thus BF played an important role in enhancing the properties of PLA. The tension–tension fatigue tests indicated that the triblock copolymers could exhibit an improved fatigue life; the fatigue lives of B1 and B2 were much lower than that of group C. C4 exhibited synergistic mechanical properties with high extensibility based on the comprehensive analyses of the tensile tests, fatigue tests, and the failure mechanism.Substantial interfacial compatibility and excellent mixing homogeneity were observed between BF, PBAT, and PLA; the scheme of the polymerization and the failure mechanism of the polymers support these observations. This research deduced important connections between the morphology, crystallinity, and mechanical properties of PLA ternary polymers with triblock copolymers, which can be utilized to reinforce and toughen PLA and optimize its degradation.The results described herein established that the materials yielded by this novel processing method exhibited similar properties to those of the original polymers. Thus, one can use BF to increase the strength of PLA, and triblock copolymers can facilitate the tension–tension fatigue properties of the PLA/BF polymers. Excellent mechanical properties expand the toughness and strength of PLA, and thus its applications such as to biodegradable battery covers.

## Figures and Tables

**Figure 1 polymers-14-02734-f001:**
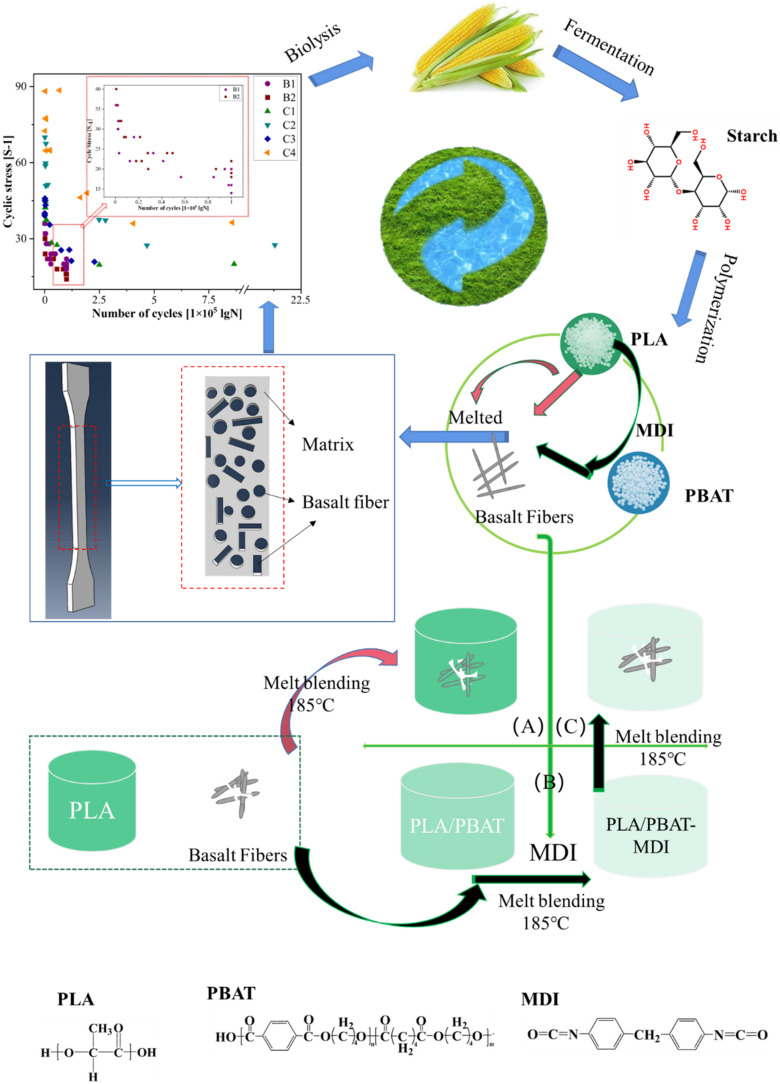
Scheme of the high-performance, biodegradable, bio-based composites, with (A), (B) and (C) are the three experimental groups in this paper.

**Figure 2 polymers-14-02734-f002:**
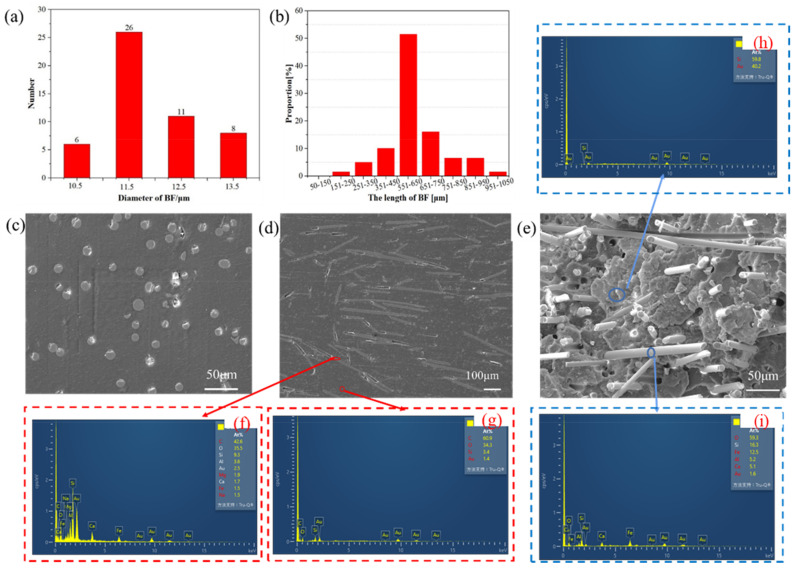
The structural characterizations of the PLA-based binary and ternary blends: (**a**) diameters from 10.5–13.5 μm, (**b**) length of the BF, (**c**) the surface SEM micrographs of C2, and (**d**) the cross-sectional SEM micrograph of C2. (**f**,**g**) are the EDS from the SEM micrograph of (**d**), (**e**) is the tensile fracture morphology of C2, (**h**,**i**) are the EDS from the SEM micrograph of (**e**).

**Figure 3 polymers-14-02734-f003:**
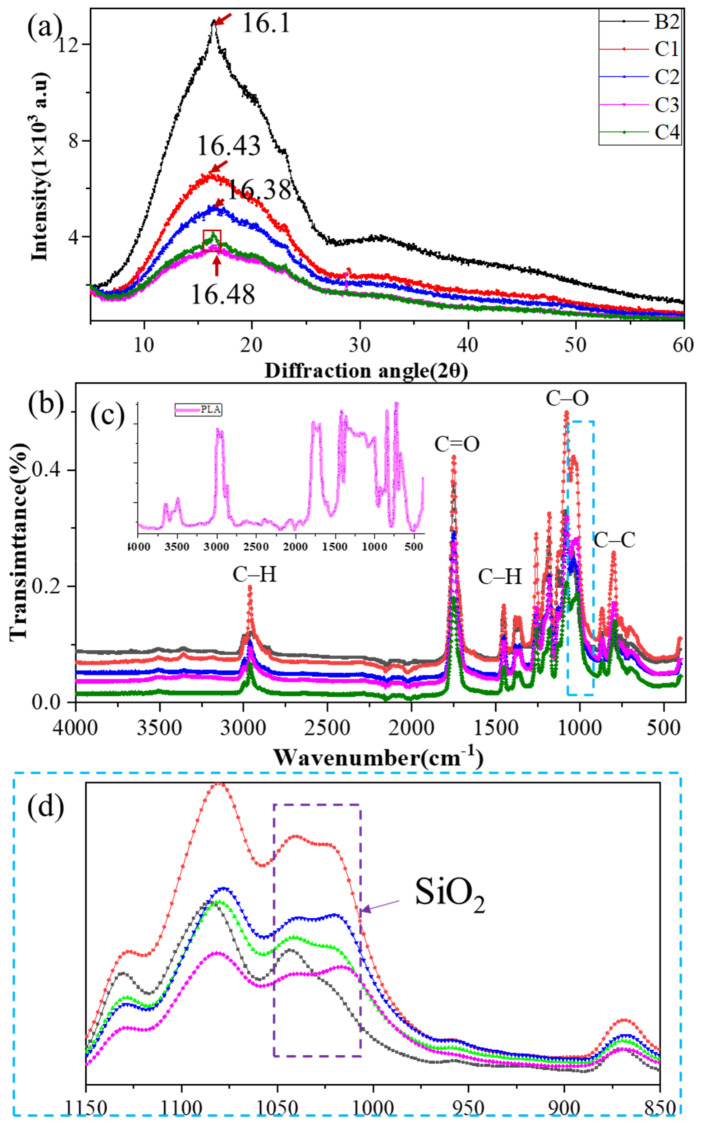
The structural characterizations of group C: (**a**) WAXS, (**b**) FTIR, (**c**) is the FTIR of PLA and (**d**) is the enlarged view of the blue area in (**b**).

**Figure 4 polymers-14-02734-f004:**
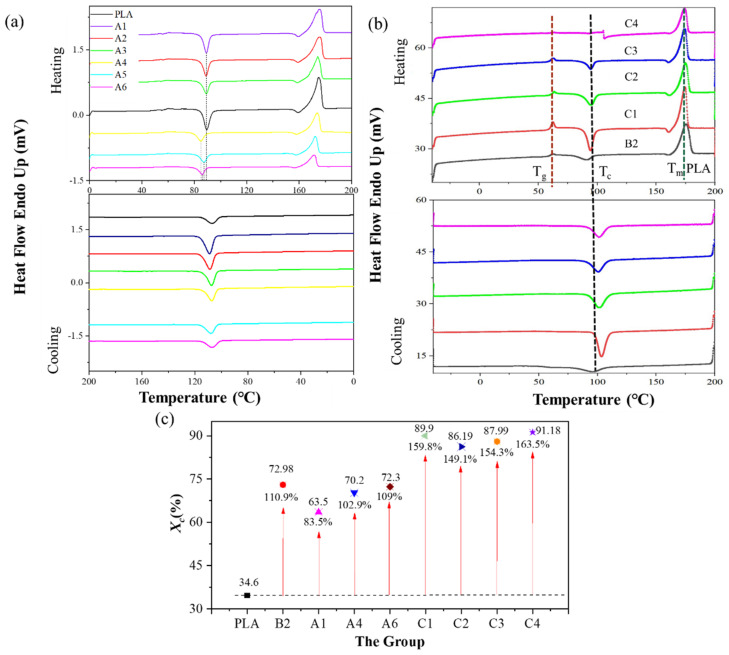
The thermal behavior of the polymers: (**a**) the DSC curves of group A, (**b**) the DSC curves of group C, and (**c**) *X_c_* (percent).

**Figure 5 polymers-14-02734-f005:**
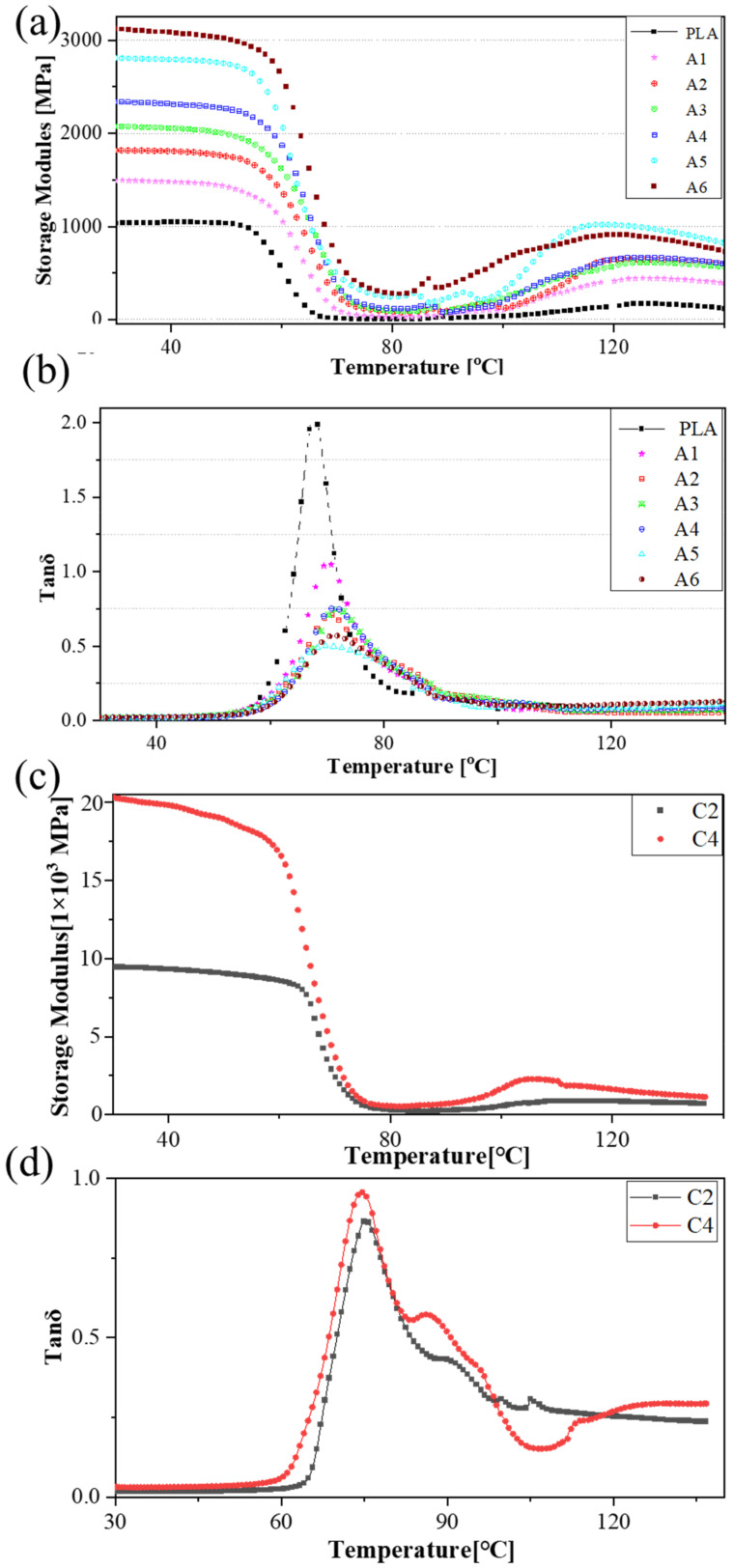
The mechanical properties of the polymers with a focus on the thermal endurance properties: (**a**) the storage modulus and (**b**) tanδ curves of group A, and (**c**) the storage modulus and (**d**) tanδ curves of C2 and C4.

**Figure 6 polymers-14-02734-f006:**
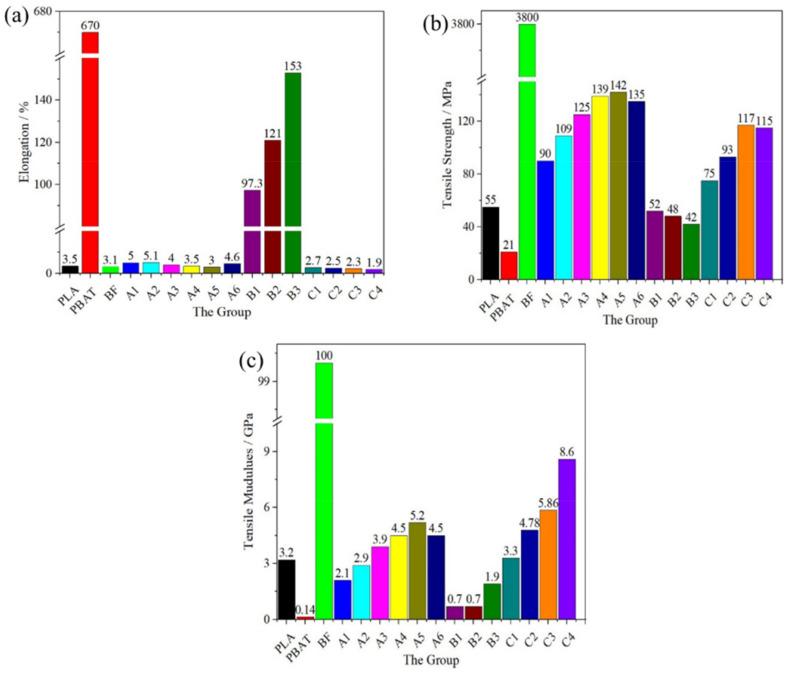
The mechanical properties of the polymers with a focus on the tensile properties: (**a**) elongation at break, (**b**) tensile strength, and (**c**) tensile modulus.

**Figure 7 polymers-14-02734-f007:**
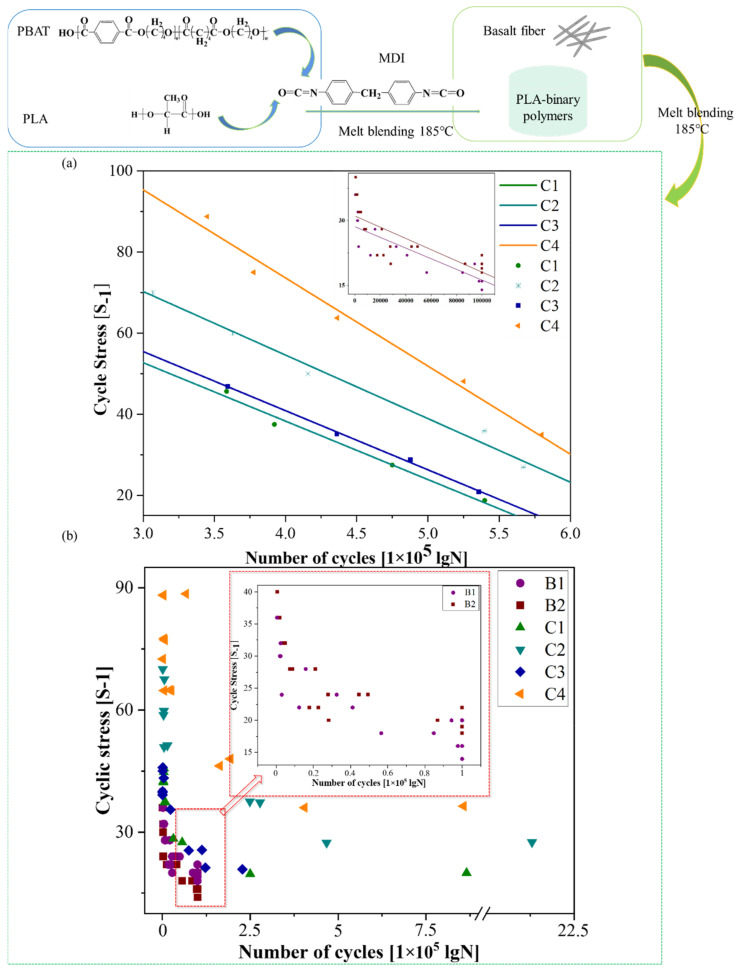
The mechanical properties of the PLA-based binary and ternary blends with a focus on the tension–tension fatigue properties: (**a**) *S*–*N* curves and (**b**) fatigue point.

**Figure 8 polymers-14-02734-f008:**
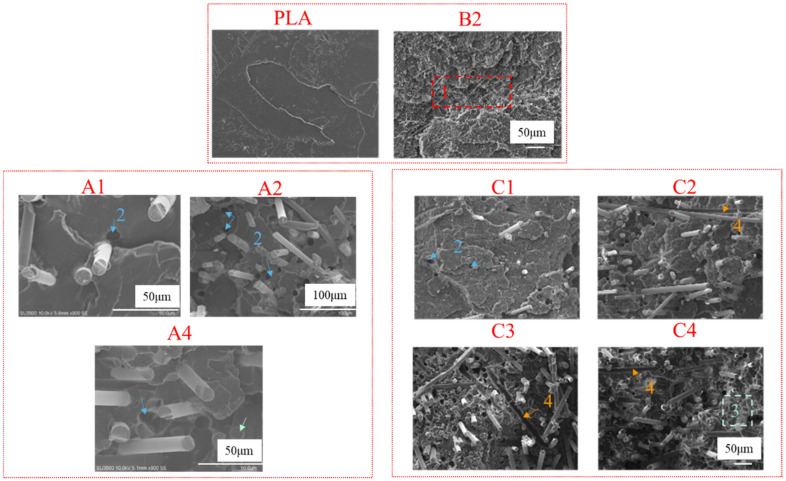
The tensile fracture morphology of PLA, groups A (A1, A2, A4), B2, and group C (C1-C4), where the phenomenon of area 1 was craze, area 2 was fiber pull-out, area 3 was fiber fracture, and area 4 was fiber debonding.

**Figure 9 polymers-14-02734-f009:**
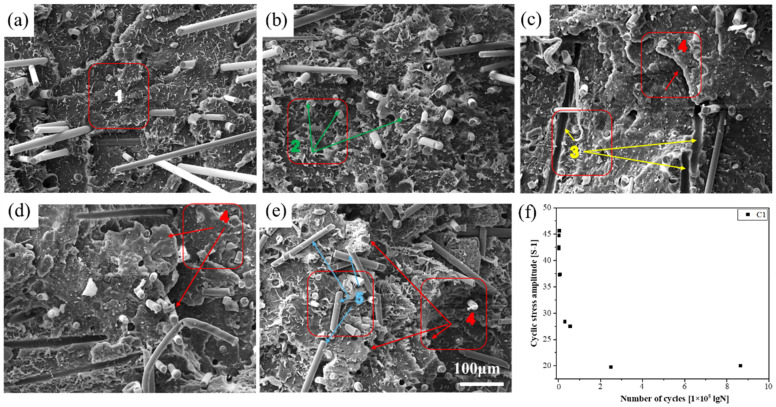
The fracture morphology and fatigue point of C1 at the following stress levels: (**a**) 30%, (**b**) 40%, (**c**) 55%, (**d**) 65%, and (**e**) 75% (area 1, craze; area 2, fiber fracture; area 3, fiber pull-out; area 4, layered peeling; and area 5, fractured BF); and (**f**) fatigue point of C1.

**Figure 10 polymers-14-02734-f010:**
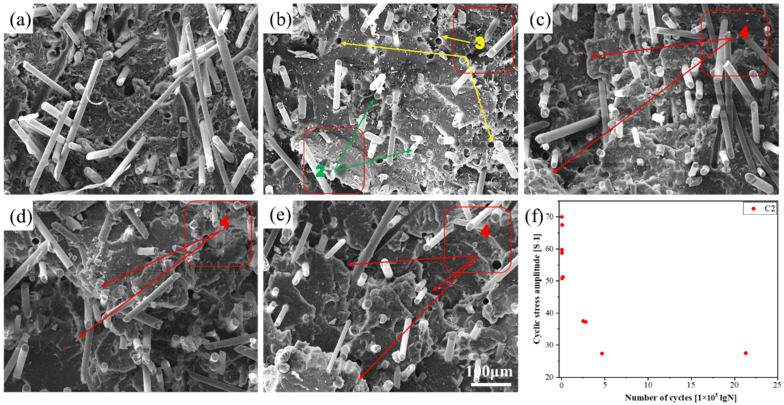
The fracture morphology and fatigue point of C2 at the following stress levels: (**a**) 30%, (**b**) 40%, (**c**) 55%, (**d**) 65%, and (**e**) 75%; and (**f**) fatigue point of C2 (area 2, fiber fracture; area 3, fiber pull-out; and area 4, layered peeling).

**Figure 11 polymers-14-02734-f011:**
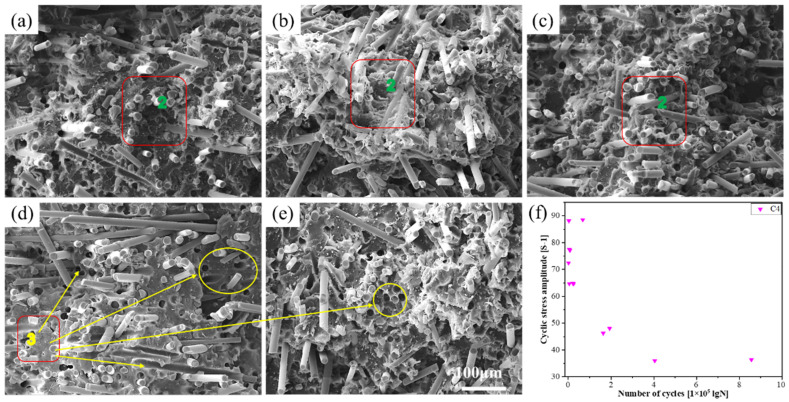
The fracture morphology and fatigue point of C4 at the following stress levels: (**a**) 30%, (**b**) 40%, (**c**) 55%, (**d**) 65%, and (**e**) 75%; and (**f**) fatigue point of C4 (area 2, fiber fracture; and area 3, fiber pull-out).

**Figure 12 polymers-14-02734-f012:**
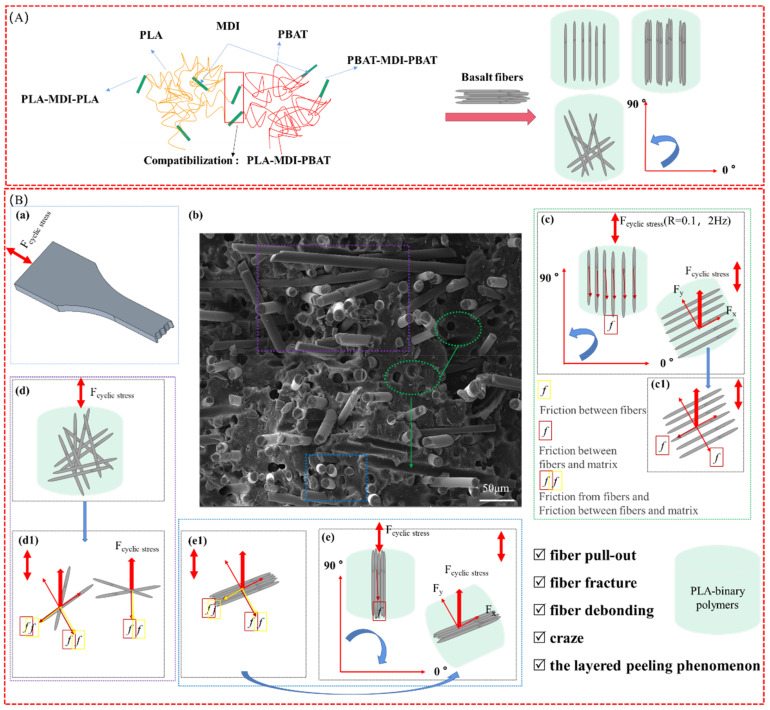
The polymerization schemes and fatigue failure mechanism of the polymers. (**A**) Polymerization schemes, (**B**) Fatigue failure mechanism: (**a**) scheme of the fatigue test; (**b**) fracture morphology of the PLA ternary polymers, with the phenomenon of fiber pull-out, fiber fracture, fiber debonding, craze and the layered peeling; (**c**–**e**) are the force status between BF and the matrix; **c1**–**e1** are the microscopic stress decomposition diagrams corresponding to (**c**–**e**).

**Table 1 polymers-14-02734-t001:** The properties of PLA, PBAT, and BF.

Materials	Density (g/cm^3^)	Tensile Strength (MPa)	Tensile Modulus (GPa)	Elongation/%
PLA	1.85	55	3.2	3.5
PBAT	1.22	21	0.14	670
BF	2.5–2.65	3200–4200	91–110	3.1

## Data Availability

The raw/processed data required to reproduce these findings are available upon request from the authors.

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
