# Peer review of "Triblock Copolymer Compatibilizers for Enhancing the Mechanical Properties of a Renewable Bio-Polymer"

_polymers, 2022, doi:10.3390/polym14132734_

Round 1
Reviewer 1 Report
Thank you for submitting your paper. The work done here draws attention to a significant subject of improving performance of biopolymers. I have found the paper to be interesting. However, several issues need to be addressed properly before the paper is being considered for publication. My comments including major and minor concerns are given below:
Please consider reviewing the abstract and highlight the novelty, major findings, and conclusions. I suggest reorganizing the abstract, highlighting the novelties introduced. The abstract should contain answers to the following questions:
What problem was studied and why is it important?
What methods were used?
What conclusions can be drawn from the results? (Please provide specific results and not generic ones).
The abstract must be improved. It should be expanded. Please use numbers or % terms to clearly shows us the results in your experimental work.
Please consider reporting on studies related to your work from mdpi journals.
The introduction is too short and authors need to expand it, mention in details past studies similar to this work, what they did and what were their main findings and critically evaluate their results against each then mention how does your work brings new knowledge and difference to the field.
Authors should remove all bulk citations unless they are given full credit.
The authors should remove all bulk citations, unless given full credit afterwards. The authors should check for this issue elsewhere in the manuscript.
Combine any small paragraphs of 5 lines or less with other paragraphs to improve the readability of the manuscript.
2. Experimental change to Materials and methods
“which exhibited good interfacial compatibility.” How did the authors verify that?
“Damage to the surface and cross sections of the polymers were attributable to polishing and inevitable.” So does the authors think this might have an impact on the performance of the samples?
“thus, the difference might be due to SiO2” how? The authors need to elaborate further
“Adding BF and PBAT will not destroy the basic skeleton of PLA, yet will enhance the performance.” Enhance what exactly? The authors need to complete their sentences and elaborate further when stating some findings.
Parts of page 6 can be moved to section 2 as it describes how things were measured and not reporting on results of the study.
“The heat energy released when the material crystallized gradually decreased; the uncrystallized part of the material decreased with increasing BF content, and thus BF played an auxiliary role in the crystallization behavior.” Why? The authors need to explain these observations in some detail.
“The tensile strength of the polymers gradually increased with increasing content of BF, this result indicates that BF played an important role in enhancing the properties of PLA.” How did it play this role, elaborate further and support with references.
Check page 15 some issues with letter f
The results are merely described and is limited to comparing the experimental observation and describing results. The authors are encouraged to include a more detailed results and discussion section and critically discuss the observations from this investigation with existing literature.
Conclusion can be expanded or perhaps consider using bullet points (1-2 bullet points) from each of the subsections.
Reviewer 2 Report
The paper seeks to introduce an approach ‘’Triblock Copolymer Compatibilizers for Enhancing the Mechanical Properties of a Renewable Biopolymers”. However, the authors should consider improving upon the quality to further highlight and emphasis.
1. Based on the understanding of what should be included in the abstract, consider adding one or two lines highlighting the significance of the study
2. Put space between each value and its corresponding units. Consider spacing between the values and their percentage units.
3. The introduction needs to be improved by relating to the mechanics of the studied materials and their characteristics. The references to be included are: 10.1007/s10853-022-06994-3 and 10.1016/j.compstruct.2021.114698.
4. The first figure was named “scheme 1” instead of “figure 1”. Accordingly, correct it.
5. Again, the figure under the biolysis of your scheme 1 has texts which blur and hard to read. Consider increasing either the font size or pasting it as text to increase its visibility.
6. One standard of spelling should be adopted. You used ‘figure’ and ‘fig.’ in the manuscript. Consider adopting one style (in full or abbreviated).
7. In figure 7, you indicated magnification of some of the images whiles others were not labeled. Any reasons for doing that and if not indicate all. The same thing applies to figure 8.
8. Increase the font sizes of all the figures like 14 and make them uniform.
9. In section 3.4, paragraph 4, the word f inside the box is misplaced and not in line. Place it well in the text.
10. List in a table form the physical and the chemical properties of the materials used for the experimental work.
11. In what proportions were each of these materials blended.
12. Was there any laid down procedures followed in minimizing the human errors when preparing the materials for the experiments?
13. What is the accelerating voltage applied and can you show the scale bar for the SEM analyses?
14. What can we infer from the SEM, and FT-IR analyses? In other words, what are you trying to achieve or how is it relevant to your work?
15. Do the figures represent an average value? If so, how many samples were tested?
Round 2
Reviewer 1 Report
All questions answered and paper can be accepted. Congratulations to the authors